# Revealing Genetic Differences in Fiber Elongation between the Offspring of Sea Island Cotton and Upland Cotton Backcross Populations Based on Transcriptome and Weighted Gene Coexpression Networks

**DOI:** 10.3390/genes13060954

**Published:** 2022-05-26

**Authors:** Shengmei Li, Shiwei Geng, Bo Pang, Jieyin Zhao, Yajie Huang, Cun Rui, Jinxin Cui, Yang Jiao, Ru Zhang, Wenwei Gao

**Affiliations:** College of Agriculture, Xinjiang Agricultural University, Urumqi 830052, China; lishengmei20201231@163.com (S.L.); gengshiwei20201231@163.com (S.G.); 13894100819@163.com (B.P.); cottonzjy@126.com (J.Z.); huang540814731@163.com (Y.H.); rc1728891526@163.com (C.R.); cuijinxin2022@163.com (J.C.); 18599052306@163.com (Y.J.); zr13094012400@163.com (R.Z.)

**Keywords:** *G. hirsutum* × *G. barbadense*, fiber elongation, backcross population, RNA-seq, STEM, WGCNA

## Abstract

Fiber length is an important indicator of cotton fiber quality, and the time and rate of cotton fiber cell elongation are key factors in determining the fiber length of mature cotton. To gain insight into the differences in fiber elongation mechanisms in the offspring of backcross populations of Sea Island cotton Xinhai 16 and land cotton Line 9, we selected two groups with significant differences in fiber length (long-fiber group L and short-fiber group S) at different fiber development stages 0, 5, 10 and 15 days post-anthesis (DPA) for transcriptome comparison. A total of 171.74 Gb of clean data was obtained by RNA-seq, and eight genes were randomly selected for qPCR validation. Data analysis identified 6055 differentially expressed genes (DEGs) between two groups of fibers, L and S, in four developmental periods, and gene ontology (GO) term analysis revealed that these DEGs were associated mainly with microtubule driving, reactive oxygen species, plant cell wall biosynthesis, and glycosyl compound hydrolase activity. Kyoto encyclopedia of genes and genomes (KEGG) pathway analysis indicated that plant hormone signaling, mitogen-activated protein kinase (MAPK) signaling, and starch and sucrose metabolism pathways were associated with fiber elongation. Subsequently, a sustained upregulation expression pattern, profile 19, was identified and analyzed using short time-series expression miner (STEM). An analysis of the weighted gene coexpression network module uncovered 21 genes closely related to fiber development, mainly involved in functions such as cell wall relaxation, microtubule formation, and cytoskeletal structure of the cell wall. This study helps to enhance the understanding of the Sea Island–Upland backcross population and identifies key genes for cotton fiber development, and these findings will provide a basis for future research on the molecular mechanisms of fiber length formation in cotton populations.

## 1. Introduction

Cotton is not only economically important but also an important fiber crop in the globalized textile industry [1]. The cotton genus includes 45 diploid cotton species and seven allotetraploid cotton species [2], and only four cotton species can be used for textile fiber production, namely, *Gossypium herbaceum* (diploid), *arboretum* (diploid), *hirsutum* (tetraploid), and *barbadense* (tetraploid) [3]. Tetraploid cotton may be the result of a cross between diploid *raimondii* (D5) and the possibly extinct diploid A0 [4,5,6,7,8]. Upland cotton production is high; thus, it occupies an important position in world production, whereas Sea Island cotton has a limited planting area and produces good fiber quality but lower yields [9,10,11,12]. With the globalization of the cotton trade and the development of the textile industry, the need for cotton farmers to increase their income, and the problem of land competition between grain and cotton, it is of great theoretical and practical importance to improve cotton fiber quality by conventional breeding methods such as crosses, backcrosses, and intercrosses and by studying cotton fiber growth and development at the molecular level [13,14,15,16,17,18].

Cotton fibers can be used to study cell elongation, cell wall thickening, and cellulose synthesis, which are formed by the protrusion of individual cellular trichomes from the outer epidermis of ovules [19,20,21]. Cotton fiber development consists of four main overlapping processes [22]: fiber differentiation initiation [23] (0 to 3 DPA), fiber cell elongation [24] (1 to 20 DPA), fiber secondary wall thickening [25,26] (16 to 40 DPA), and fiber dehydration maturation [27] (40 to 50 DPA). Each of these four stages has its own characteristics, and there are no strictly distinguishable boundaries. The state of the fiber at different stages of development can affect the final quality characteristics, such as fiber length (FL), macronaire (FM), and fiber strength (FS). [28]. Cotton fibers consist mostly of cellulose, and approximately one-third of the epidermal cells can differentiate into spinnable fibers [29,30], while the length of cell differentiation affects the length of mature fibers, coordinated by different regulatory mechanisms.

Cotton fiber quality is an important agronomic trait; therefore, significant efforts have been made to study its biological mechanisms [31,32,33,34,35]. In recent years, several key genes involved in fiber development have been reported [36,37,38,39,40], which have greatly contributed to the study of fiber development mechanisms, and studies have shown that hormones, functional genes, and transcription factors form a complex regulatory network that coordinates the dynamic fiber development process [3]. Many factors affect cotton fiber development, such as regulation by plant hormones such as auxin [41], ethylene [42,43], abscisic acid [44,45], brassinosteroids [46], cytokinins [47] and gibberellins [48]. In addition, lipids [49], cellulose [50,51,52], cell wall proteins [53], cytoskeletal proteins, and their binding proteins [54], and reactive oxygen species [55,56] have also been demonstrated. In addition, a series of genes, including transcription factors [57], glucose metabolism [58], and secondary metabolites [59,60,61], also plays key roles in development.

Among the plants, some candidate genes or intervals have been discovered by quantitative trait locus (QTL) mapping and association analysis [62,63,64]. However, cotton QTL localization faces problems such as large localization intervals, small recombination exchange probability, and high density of genetic linkage maps, which are still challenging for mining fiber development-related genes. Fortunately, as traditional sequencing technologies are refined and updated, RNA-seq provides a suitable mining process that has been widely applied to transcriptome studies in several species, including Arabidopsis [65], poplar [66], soybean [67], rice [68], wheat [69], cotton [70,71,72,73], and maize [74]. By comparing the transcriptomes of different cotton fiber development samples, the majority of genes responsive to cell development can be identified quickly and efficiently [75,76]. However, few studies have focused on the use of extreme cotton fiber material in backcross populations to study elongation. In addition, Weighted gene coexpression network analysis (WGCNA) is very quick and efficient for mining functionally relevant genes in coexpression modules, and it has been widely used in a variety of crops to mine the corresponding candidate genes [77,78,79,80].

To better elucidate the genetic basis of fiber length, we constructed backcross populations of Sea Island cotton and Upland cotton. In this study, we selected long-fiber offspring L and short-fiber offspring S at 0 to 15 DPA for transcriptome sequencing, revealed possible response pathways for offspring cotton fiber development by comparing the DEGs generated from transcriptome analysis with WGCNA, and identified some key candidate genes that may have a strong influence on cotton fiber elongation. This study provides insights and evidence to further elucidate the molecular basis of fiber length traits and provides guidance for the future breeding of high-quality cotton varieties.

## 2. Materials and Methods

### 2.1. Plant Materials and Sample Collection

In this study, six offsprings (HL-9, HL-34, HL-62, HL-78, HL-159, and HL-194) of BC_4_F_2:6_ planted on 21 April 2020, at the cotton breeding base of Xinjiang Agricultural University, 144 Mission, Shawan City, Xinjiang Province, were used. Because of the focus on population fiber length, for three years, we selected materials with stable fiber lengths but with large differences between offspring (Table 1). HL-9, HL-62 and HL-194 were always greater than 30 mm of FL, so these three offspring were defined as long-fiber group L; the FL of HL-34, HL-78, and HL-159 was always less than 26 mm. For a randomized zonal design with a six-row planted pattern with two rows per offspring (2 m row length, 0.1 m plant spaced, 0.7 m row spaced, 2.35 m sowed width, three replications (50 plants per replicate)), drip fertilization beneath mulched film was used for plant growth, and conventional field management was used.

### 2.2. Determination of Fiber Quality Phenotypes

In early October, 20 cotton bolls with natural bolls were harvested manually (one boll per plant), and approximately 12 g of mature fiber was weighed after ginning and sent to the Cotton Quality Supervision and Inspection Center (HVI1000) of Shihezi Academy of Agricultural Reclamation Sciences to determine fiber quality with three biological replicates [81]. The *t* test was performed using SPSS 26.0.

### 2.3. RNA Extraction, Library Construction, and Sequencing

Sampling began on 5 July 2020, with the initial date of collection being the day of flowering, labeled 0 DPA, and 36 developing cotton bolls from six offspring labeled 0 DPA, 5 DPA, 10 DPA, and 15 DPA were harvested daily at 10:00 a.m. after flowering of each replicate; then, they were immediately placed in foam boxes with ice packs [82]. Three replicates were labeled and sampled for a total of 108 bolls. Fiber samples were placed in liquid nitrogen storage for subsequent RNA-seq and qRT–PCR validation.

Total RNA was extracted from frozen cotton fiber tissue using TRIzol reagent (Tiangen DP411, Beijing, China). RNA degradation and contamination were examined by 1% agarose gel electrophoresis, and concentration detection was performed using a Nanodrop 2000 (Thermo Fisher Nanodrop 2000, Shanghai, China) before calibration using an Agilent 2100 (Platinum Elmer LabChip GX, Beijing, China) for RNA integrity. The RNA concentration was calibrated according to Qubit quantification (Life Technologies, Beijing, China). To minimize experimental error, after calibration, RNA samples extracted from the offspring with long fibers and the offspring with short fibers at each stage were prepared at the same concentration and volume. The Sea Island cotton and Upland cotton backcross populations are more consistent in the research background. In order to reveal the overall genetic characteristics, we chose samples that were were mixed to generate one sample L and one sample S with 3 biological replicates each, which was designed to highlight the differences between the 2 groups of cotton and between the different stages fiber development stages (Appendix A). The samples were calibrated again and used for RNA-seq, subject to same volumes, quantity, an OD260/280 between 1.8 and 2.2, an OD260/230 between 1.8 and 2.2, a RIN value ≥7, and a baseline judgment in combination with GX assay before using high-quality RNA samples for library construction.

First, mRNA Capture Beads were added to the prepared total RNA samples and mixed well and incubated at 65 °C to denature the RNA, followed by binding of mRNA to Oligo (dT) beads for mRNA purification and fragmentation. Second, the first strand of the cDNA in question was synthesized with six-base random hexamers, and then the corresponding buffers, dNTPs, RNase H_2_O, and DNA polymerase I, were added to ensure the synthesis of the second strand of cDNA and product purification. End repair was performed with 3’ plus A to connect the sequencing junction; immediately afterward, the purification of ligation products and fragment size selection were performed with AMPure XP beads; the final cDNA library was attained after amplification and purification. The libraries were quality checked by the Qsep-400 method. The Illumina HiSeq NovaSeq 6000 platform was used for sequencing (Illumina, San Diego, CA, USA). Four developmental stages were prepared using mRNA isolated from 2 groups of cotton fibers from L and S. A total of 24 cDNA libraries were studied (Appendix A).

### 2.4. Data and Differential Gene Expression Analysis

Raw reads were obtained using Illumina. First, fastp software [83] was used to filter and quality control the data to obtain clean reads while calculating the Q30 and GC content. Then, the clean reads were mapped to the *Gossypium hirsutum* reference genome (TM_1_v2.1) using HISAT2 software [84] and quantified using StringTie software [85] based on Fragments per kilobase of transcript per million fragments mapped (FPKM) to estimate the gene expression level of each transcript [86]. Using the EdgeR package [87], genes with a fold change ≥ 1.5 and FDR < 0.05 were designated DEGs [88]. The false discovery rate (FDR) was calculated by adjusting the *p* value with multiple corrections for the false discovery rate [89]. In addition, Pearson correlation coefficient (PCC) analysis was completed using the R3.6.0 language, and Principal component analysis (PCA), GO analysis, and KEGG enrichment analysis were completed using the online data analysis platform BMKCloud tool (www.Biocloud.net, accessed on 26 February 2021). TBtools was used to plot Venn diagrams and heatmaps of DEGs [90]. Trend analysis of DEGs was completed using STEM software [91]. Parameter settings: the maximum unit change of the model profile between time points was 1, the maximum output profile was 20, and the minimum multiplicative change rate of DEG was greater than 1.5.

### 2.5. WGCNA Construction and Hub-Gene Screening

A coexpression network of cotton fiber DEGs was established using the “WGCNA” R package [92] using the dynamic cut synthesis module. In a scale-free weighted gene network, a node corresponds to a DEG, and an edge is determined by the similar expression profile of paired genes calculated by Pearson’s correlation. Therefore, we chose an optimal soft threshold β = 23 to construct the network based on the adjacency matrix. The coexpression network was visualized using Cytoscape 3.9.1 software. The CDSs and protein sequences of 21 hub genes in Upland and Sea Island cotton were extracted from the cotton genome library (https://cottonfgd.org/, accessed on 30 October 2021) and BLAST using sequenceserver [93].

### 2.6. qRT–PCR Validated Transcriptome Sequencing

Primer BLAST (https://www.ncbi.nlm.nih.gov/tools/, accessed on 7 May 2021) was used to design gene-specific primers. The FastKing RT Kit (Tiangen KR116, Beijing, China) was used for reverse transcription. The PerfectStart Green qPCR SuperMix (TransGen BiotechAQ601, Beijing, China) kit was used for fluorescence quantitative amplification in a volume of 20 µL. The reaction mixture contained 10 µL of 2× PerfectStart Green SuperMix, 2 µL of template cDNA, 0.4 µL of F Primer (10 µM), 0.4 µL of R Primer (10 µM), 0.4 µL of passive reference dye (50×), and 6.8 µL of nuclease-free water. The qRT–PCR procedure was a two-step method, using ABI 7500 Fast to complete the amplification, with the following procedure: 30 s of predenaturation at 94 °C followed by 40 cycles of 5 s of denaturation at 94 °C and 34 s of fluorescence signal acquisition at 60 °C. The experiment was performed with three replicates, the internal reference gene was UBQ7, and the relative gene expression levels were calculated by the 2^−∆∆Ct^ method [94] with the primer sequences listed in Appendix A.

## 3. Results

### 3.1. Fiber Phenotypes of Extreme Offspring of Fiber Length

After fiber maturation, the fiber length of six offspring was measured, and these values were 30.40 mm, 30.16 mm, and 31.62 mm and 25.60 mm, 24.68 mm, and 25.86 mm (Table 1). The differences between L and S were highly significant according to the results of the *t* test. Parental Line 9 also differed very significantly by 6.35 mm in FL compared to Xinhai 16. In addition to FL, four other fiber characteristics (fiber strength, macronaire, uniformity, and elongation) were compared. Overall, the differences in the four fiber characteristics between the extreme offspring were also more pronounced (Table 1). The differences in fiber length were more pronounced and stable compared to the other traits; therefore, these six offspring may constitute a good model to reveal the molecular mechanisms controlling the differences in fiber elongation in the backcross populations of *G. hirsutum* × *G. barbadense*.

### 3.2. Overall Analysis of the Transcriptome during Cotton Fiber Development

To explore the important genes related to cotton fiber elongation development in the progeny of Sea Island–Upland backcrosses, 24 cDNA libraries were constructed and sequenced for the progeny with significant differences in fiber length in this assay. Among them, each sample of clean data reached more than 5.87 Gb, and a total of 171.74 Gb of clean data was obtained. The alignment efficiencies with TM-1 were all over 92.02%, Q30 over 93.70%, and GC content over 43.60% (Appendix A). All these results indicated the high quality of RNA-seq in this study for subsequent analysis.

Based on the alignment results, 11,417 new cotton genes were observed, 7022 of which were functionally annotated (Appendix A). Gene expression levels were measured by normalized FPKM values, and the PCC method was used to detect the correlations between all samples. The overall correlation between the three biological replicates at the same developmental stage was high in the two groups of cotton lines, and sample clustering analysis showed similar expression patterns between L and S at the same developmental stage and different expression patterns at different developmental stages (Figure 1A), which indicated that our experimental manipulation was competent. In the three subsequent developmental stages of the long-fiber group (L5, L10, and L15) and the short-fiber group (S5, S10, and S15), the cotton fiber initiation stages L0 and S0 were less correlated with the other stages (Figure 1A). These results suggest that the gene expression pattern changes dramatically with fiber development.

PCA of the above expressed genes revealed that the contribution of the two extracted PCs amounted to 93.3%, of which PC1 explained 85% of the results and could clearly distinguish the differences in fiber development days, PC2 explained 8.3% of the results and could distinguish cotton lines with different fiber lengths, and the differences between developmental stages were greater than the differences between materials (Figure 1B). In addition, samples from the two groups of cotton lines at the same time period of fiber development can be aggregated together, especially at fiber development 0 DPA, which could be related to the lack of separation of fiber and ovule. Since different samples at the same developmental stage have similar expressed genes between them, PCA indirectly proved the reliability of transcriptome data.

### 3.3. Analysis of Differentially Expressed Genes (DEGs)

To confirm the reliability of our RNA-seq, eight genes were randomly selected for qRT–PCR experiments and were found to be in general agreement with the transcriptome expression profile expression trends, proving that the transcriptome data were reliable for the next step of analysis (Figure 2). Four stages of fiber development between L and S, namely, L0 vs. S0, L5 vs. S5, L10 vs. S10, and L15 vs. S15, were found to contain 1813, 2621, 1603, and 1472 DEGs, respectively (Figure 3A and Appendix A). In different developmental stages of the same material, L0 vs. L5, L5. vs. L10, and L10. vs. L15 contained 22537, 10999, and 15869 DEGs, respectively, and S0 vs. S5, S5 vs. S10, and S10 vs. S15 contained 20241, 14308, and 15196 DEGs, respectively (Figure 3A and Appendix A). Although the number of DEGs at different developmental periods of the same material was much greater than the number of DEGs between materials, the number of DEGs at one period was basically the same for both materials. These results suggest that DEGs at different developmental stages do change dramatically, which is consistent with our PCA results (Figure 1B).

To further accurately reveal the relationship between DEGs and fiber elongation, 6055 DEGs between materials were finally selected, and GO-term enrichment analysis was used to determine the functional roles (Figure 3B and Appendix A). The GO-enriched DEG categories in the bioprocessome were mainly associated with microtubule-based movement and reactive oxygen and hydrogen peroxide responses. The GO-enriched DEG categories in the cellular fraction were associated mainly with nucleosomes, MCM complexes and plant cell wall biosynthesis. The GO-enriched DEG categories in molecular functions were associated mainly with hydrolase activity, hydrolysis of O-glycosyl compounds, microtubule motility activity, and protein complexes (Figure 3B). We also mapped DEGs to the KEGG database for pathway enrichment analysis, and the KEGG-annotated DEGs were classified into different categories (Figure 3C and Appendix A), associated mainly with plant hormone signaling, plant MAPK signaling pathways, starch and sucrose metabolism, and phytopathogen interactions. In addition, DEGs were significantly enriched in protein processing, amino acid biosynthesis, phenylpropane biosynthesis, and carbon metabolism pathways (Figure 3C).

### 3.4. Analysis of Gene Expression Patterns

We analyzed the expression patterns of 32425 and 33328 DEGs in the long-fiber group L and short-fiber group S, respectively, at four developmental stages (Figure 4A and Appendix A). The results showed that, in L, the DEGs were significantly divided into six profiles, where most DEGs were in profile 12 (9576 genes, 29.53% of the total DEGs), followed by profile 19 (7086 genes, 21.85%) and profile 18 (6514 genes, 20.08%) (Figure 5A L-profile). In S, the DEGs were significantly divided into seven profiles (Figure 5A S-profile), and a similar trend of consistently upregulated expression (profile 19) was found in L and S (Figure 4A). It has been reported that elongation-related genes are highly expressed during the fiber elongation stage and less expressed during early fiber development [73]. These findings coincide with the profile 19 expression pattern, suggesting that the DEGs of profile 19 may be involved in fiber elongation.

L-profile 19 had 4740 more DEGs than S-profile 19 (7086 vs. 2346) (Figure 4A), and GO enrichment analysis of profile 19 was performed (Appendix A). In S, profile 19 was enriched in actin cytoskeleton organization (GO: 0030036, 8.47 × 10^−1^), cellular macromolecule metabolic process (GO: 0044260, 9.67 × 10^−1^), signal transduction (GO: 0007165, 9.89 × 10^−1^), and microtubule binding (GO: 0008017, 5.77 × 10^−2^) (Figure 4B S-profile 19). In L, profile 19 was broadly consistent with the enrichment results in S, except for some special GOs, such as carbohydrate biosynthesis and metabolic processes (GO: 1901137, 9.75 × 10^−1^ and GO: 1901135, 9.99 × 10^−1^) and glycogen biosynthetic process (GO: 0005978, 8.97 × 10^−1^) (Figure 4B L-profile 19).

The previous results suggested that the genes in profile 19 might be associated with fiber elongation. Therefore, further analysis of the DEGs of L and S in fiber development at 5 DPA, 10 DPA, and 15 DPA (Figure 3A, 1668 + 953, 1018 + 585, and 937 + 535 DEGs) and genes in L-profile 19 and S-profile 19 (Figure 4A) revealed that 167 genes (103 + 58 + 6), 138 genes (91 + 38 + 5), and 144 genes (90 + 24 + 30) were differentially expressed in the fiber elongation phase (Figure 5A–C). To confirm which metabolic pathways these 167, 138 and 144 DEGs are involved in, KEGG enrichment analysis was performed (Figure 5D–F and Appendix A). Five KEGG classifications were obtained for 49 (Figure 5D), 45 (Figure 5E), and 56 (Figure 5F) DEGs at 5 to 15 DPA, respectively, with similar metabolic pathways. However, the number of glycan biosynthesis and metabolism-related DEGS increased as the number of days of fiber development increased. Subsequently, 129 DEGs were homologously compared and functionally annotated in the Arabidopsis database (Appendix A). This included genes encoding 3 glycosyltransferases, 6 hydrolases, 2 peroxidases and oxidoreductases, 4 ABC transporter proteins, 2 transcription factors, 4 protein kinases, 4 growth hormone response factors, 10 heat shock proteins, and several genes involved in signal transduction, carbohydrate, starch, and lipid metabolism.

### 3.5. Coexpression Network Analysis and Identification of Hub Genes

To further understand the linkage between fiber development and gene expression and to mine candidate genes related to fiber elongation, 6055 DEGs were used with WGCNA to construct networks. The soft threshold of β was set as 23, the scale-free R^2^ > 0.80, and the correlation coefficients between DEGs were calculated by a dynamic shear algorithm. Clustering and module division were performed on the basis of constructing a matrix with different module colors, and finally, a total of 15 expression modules were obtained (Figure 6A). Based on the correlation results between the expression module and the two samples, five modules were highly correlated (correlation index > 0.8) with trait characteristics (Figure 6B). The MEpink module was correlated with long-fiber progeny L-fiber development at 0 DPA (Cor = 0.90, *p* = 0.002); the MEmagenta module was strongly correlated with long-fiber progeny L-fiber development at 10 DPA (Cor = 0.94, *p* = 5 × 10^−5^); the MEblack module was strongly correlated with long-fiber progeny L fiber development at 15 DPA (Cor = 0.93, *p* = 9 × 10^−4^); the MEbrown module was strongly correlated with short-fiber progeny S fiber development at 10 DPA (Cor = 0.81, *p* = 0.01); the MEblue module was strongly correlated with short-fiber progeny S fiber development at 15 DPA (Cor = 0.81, *p* = 0.01) (Figure 6B).

Four modules that were strongly associated with fiber development at 10 DPA and 15 DPA and were specifically identified were selected for the construction of gene interaction networks and hub screening. To identify the major hub genes of these modules, gene networks were visualized using Cytoscape 3.9.1. In total, 21 hub genes were identified (Figure 7 and Table 2), and in the magenta module, hub genes encoded indicator proteins, pectinases, and extension proteins (Figure 7A and Table 2). The highest pivotal gene degree in the brown module belonged to heat shock protein, followed by fibronectin synthase, transporter protein, glucosyltransferase, and homologous heterotypic domain leucine zipper protein (Figure 7B and Table 2). Key genes in the black module included GH_A03G0667 (actin), GH_D11G2156 (cell cycle protein), GH_A09G2563 (glycoside hydrolase), GH_D04G0535 (ABC transporter protein), and GH_D02G0917 (receptor protein kinase) (Figure 7C and Table 2). In the blue module, the hub genes were almost entirely interrelated: genes encoding fasciclin-like arabinogalactan protein 9, proline protein, annexin protein, receptor kinase, mitogen-activated protein kinase, and endoglucanase were identified as key genes (Figure 7D and Table 2). These proteins and enzymes may affect fiber development directly or indirectly by participating in processes such as cell wall relaxation, cell wall skeleton, hemicellulose skeleton, and microtubule organization (Table 2). In addition, the nucleic acid sequences of 19 hub genes were found by BLAST to differ between Upland cotton and Sea Island cotton, resulting in different amino acid sequences in these two cotton species (Appendix A and Appendix A).

## 4. Discussion

### 4.1. Transcriptome Sequencing of Backcross Offspring Provides New Insights to Explore the Expression Profile of the Fiber Elongation Stage

The transcriptome in a narrow sense usually refers to mRNA as the object of study, which has obvious spatial and temporal limitations and is different for different tissue cells, different growth environments, and different growth stages of the same species. Cotton fiber length is determined mainly by the primordial wall synthesis period, which is a key component of fiber quality, and the systematic mechanisms affecting fiber length and development are still poorly understood [5]. Transcriptome analysis shows that gene expression patterns as well as their functional distribution vary significantly at different periods of development [95]. To explore the molecular mechanism of the fiber elongation process, we selected six extreme offspring from Sea Island cotton and Upland cotton backcross populations and mixed samples for RNA-seq. In our study, ovule and fiber samples were collected for sequencing during initiation and elongation, which provided valuable data to reveal differences in genetic and molecular mechanisms at the mRNA level during initiation and elongation. The sequencing of ovules and fiber samples during fiber initiation and elongation provided valuable data to reveal the differences in genes and molecular mechanisms at the mRNA level during initiation and elongation. Since the maternal Line 9 and the paternal Xinhai 16 as well as the offspring showed greater differences in fiber length (Table 1), we focused on the fiber length of the six extreme offspring to reveal the molecular mechanisms controlling the differences in fiber elongation. We obtained a total of 171.74 Gb of clean data from 24 cDNA libraries, and each sample had an average of 5.87 Gb of clean data, Q30 ≥ 93.70%, and GC content ≥ 43.60% (Appendix A), which proved that the RNA-seq data in this research were reliable. The high sample correlation (>0.8) among the three biological replicates indicated that the method of mixing long-fiber offspring and short-fiber offspring at each stage at the same concentration and volume to generate one sample L and one sample S is applicable and can be used to highlight the differences between the two groups of cotton at the fiber developmental stages; this is similar to the method Qin et al. [73] used to study the difference between the long-fiber group and the short-fiber group of Upland cotton through the transcriptome. Providing a solid basis for studying the transcriptional differences in elongation.

### 4.2. DEGs Reveal Transcriptional Differences in the Elongation Stage

In general, the primary cell wall contains cellulose, hemicellulose, proteins, and various polysaccharides (pectin and xyloglucan, etc.), as well as specific structures such as the benzyl propane polymer lignin; the secondary wall contains a large amount of cellulose. In addition, there are several different types of cell wall proteins [96]. These different cell wall structural components are involved in different functions of fiber development. Fiber elongation is a dynamic process that includes a series of genes involved in cellulose synthesis, microtubules, the cell skeleton, and metabolic pathways [50,51,52,53,54,59,60,61].

Hydrogen peroxide is a reactive oxygen species (ROS) that significantly promotes fibroblast elongation in vitro [55]. Microtubules are an important part of the plant cytoskeleton and play a large role in cotton fiber elongation and cotton fiber secondary wall thickening [97]. We determined by GO enrichment analysis that DEGs between the two groups of fiber length extreme offspring were significantly correlated with microtubule-based movement or motility activity, reactive oxygen and hydrogen peroxide responses, plant cell wall biosynthesis, and glycosyl compound hydrolase activity (Figure 3B), suggesting that microtubule drive and reactive oxygen species play a role in regulating elongation. A study comparing the fibers of Upland cotton and Sea Island cotton in multiple dimensions [98] revealed that delayed genes between them are mainly involved in carbohydrate metabolism, phytohormone signaling, and starch and sucrose metabolism, which is consistent with our KEGG pathway enrichment results. In addition to being significantly enriched in the plant MAPK signaling pathway (Figure 3C), MAPK is an intracellular serine/threonine protein kinase, *AtMPK4* is involved in regulating plant development, and *GhMPK6* maintains higher phosphorylation in elongating fibers by responding to plant hormones [99].

### 4.3. The DEGs Identified by STEM May Have a Significant Effect on Fiber Length

In this study, the DEGs of L-profile 19 and S-profile 19 intersected with those produced by extreme offspring at three stages of fiber development (Figure 5A–C), and KEGG analysis of these DEGs (Figure 5D–F and Appendix A) revealed the glycan biosynthesis and metabolism-related pathways ko00510, ko00513, ko00514, ko00515, and ko00603, indicating that fiber development is a process of sugar accumulation. Subsequent functional annotation of these DEGs (Appendix A) found that compared with other hormones, auxin-responsive proteins were more significantly altered, including auxin-responsive CH3 family protein genes (GH_A11G0511) and SAUR-like auxin family genes (GH_A03G2259, GH_A03G2261, and GH_A12G0395), demonstrating that auxin regulation is important for fiber elongation [100,101,102]. A larger number of glycosyltransferases (GH_A01G0799, GH_A09G0685, and GH_D09G0620) are also involved, and the glycosyltransferase family has been reported to take part in the synthesis of xylose, pectin, and xyloglucan, which maintained the normal morphology of plant cell walls and may be associated with fiber development [103]. The upregulation of hydrolases results in the progressive degradation of related sugar, pectin, and xyloglucan molecules [104], and in our study, six DEGs were annotated as hydrolases (GH_A09G0946, GH_A11G0610, GH_A13G1937, GH_A13G2163, GH_D05G0822, and GH_D05G2339). ABC transporter proteins can transport a variety of substances across membranes and are powered by ATP hydrolysis [105]. For example, *GhWBC1* is rapidly expanded and highly expressed in fibroblasts from 5 to 9 DPA, and the overexpression in Arabidopsis results in short horned fruits [106]. Similarly, we identified four ABC transporter protein genes (GH_A10G0335, GH_D04G0535, GH_D11G3740, and GH_D11G3740) and additional protein kinase genes (GH_A11G329, GH_D06G2149, GH_D07G0144, and GH_D09G2213) and heat shock protein genes (GH_A02G0345, GH_A03G0302, and GH_A05G0923, etc.), of which HSP70 and HSP90 were found to promote fiber elongation-related genes and maintain ROS by regulating intracellular H_2_O_2_ levels, playing a positive role in cotton fiber development [107]. These results suggest that these DEGs may play a role in fiber developmental elongation.

### 4.4. The Hub Gene Identified by WGCNA May Have a Significant Effect on Fiber Length

WGCNA is suitable for complex data patterns and can be applied to studies on developmental regulation of different organs or tissue types [77,78,79], developmental regulation of the same tissue at different times [108,109], response to abiotic stress at different time points [110,111,112], and response to pathogen infestation at different time points [73,113,114]. Zhou et al. [77] used WGCNA to reveal five genes mediated by auxin and gibberellin as node genes in the regulatory network of flowering time in cotton. Jiang et al. [78] applied WGCNA to the RIL population to identify 29 pivotal genes related to cellular elongation. Li et al. [108] identified a module with the strongest correlation with chicken pectoral development by this method, in which seven pivotal genes were found and validated. The *DYNLL2* gene was found to promote myoblast differentiation. Cheng et al. [112] identified four modules closely related to low-temperature stress with the help of WGCNA and identified 936 hub genes.

This shows that the WGCNA of key genes is more mature and superior for identifying functionally relevant genes. Therefore, we used the same analysis and found that the core DEGs were divided into four modules by WGCNA, each providing different metabolic pathways related to fiber elongation. In the coexpression network in this study (Figure 7), we found that two cell wall proteins, proline-rich proteins and expansins, actin, ABC transporters, heat shock proteins, annexins, homeodomain leucine zipper proteins, fascin-like arabinogalactosides, mitogen-activated protein kinase, glycosyl hydrolase, cellulose synthase, pectin-like enzymes, and several glucose metabolism-related genes occupied important hubs at 10 DPA and 15 DPA (Table 2). Among them, GH_D08G2106 in the magenta module is annotated as a pectinase (Figure 7A and Table 2). Pectin lyase and pectin esterase are degraded, and cell wall relaxation affects their chemical properties, playing an important physiological role in fiber development [26,115]. Lv et al. [53] have shown that expansin (GH_D04G0452) is associated with fiber elongation. Phosphorylation of a heat shock protein (GH_A05G0924) in the brown module has been shown to affect the structure of the cytoskeleton [116]. The cellulose synthase superfamily contains cellulose synthase genes (CES) and cellulose synthase-like genes (CSL) [51], and the overexpression of *GhCSLD3* restores cell elongation [52]. In our study, two CSL genes (GH_D09G2382 and GH_D02G0408) were identified (Figure 7C and Table 2). This is similar to previous results [117], suggesting that cellulose synthase, expansin proteins, and glycosyl hydrolases may be potential targets for improving fiber length. The blue module GH_D12G1588 is an *FLA9* gene, and FLA is a subclass of proteins of arabinogalactan that is of interest for plant development [118]. GH_A10G2000 was annotated as annexin RJ4, which may be associated with membranes through Ca^2+^ signaling and actin assembly to regulate fiber cell elongation, and *GbAnx6*, *GhAnn3*, *GhAnn4*, and *GhAnn5* are specifically expressed during fiber elongation [96]. As mentioned above, receptor-like kinases (RLKs) on pectin are also essential for the regulation of cell growth. RLKs have been reported to be involved in SCW synthesis phase expression [119,120], but the detailed mechanism in fiber elongation remains to be studied. In our study, the LRK10L gene (GH_D11G1764) is a hub gene and is highly expressed in S. The nucleic acid sequences of 19 hub genes were found by BLAST to be different in Upland cotton and Sea Island cotton (Appendix A), leading to differences in the amino acid sequences of the two cotton varieties and possibly in turn to differences in their physicochemical properties. The next step is to verify the function of these 19 genes in fiber development by performing genetic transformation and other experiments. Studying the differences in genes correlated with fiber development between Sea Island cotton and Upland cotton will be of great significance to improve cotton fiber quality. The genes identified in this paper may be particularly good candidates for future studies on fiber length formation and for improving cotton fiber quality by using molecular breeding.

## 5. Conclusions

In this study, we performed differential analysis, enrichment analysis, expression pattern analysis, and WGCNA using RNA-Seq data on two selected groups of progeny with extreme fiber length at different stages of fiber development in Sea Island cotton and Upland cotton backcross populations. We found that hormone signal transduction, MAPK signal transduction, and starch and sucrose metabolism may play important roles in fiber elongation. Based on WGCNA, 21 hub genes were possibly related to fiber elongation. This study not only preliminarily analyzed the molecular mechanism of the fiber elongation difference between Sea Island cotton and Upland cotton but also laid a foundation for the future study of the molecular mechanism of fiber length.

## Figures and Tables

**Figure 1 genes-13-00954-f001:**
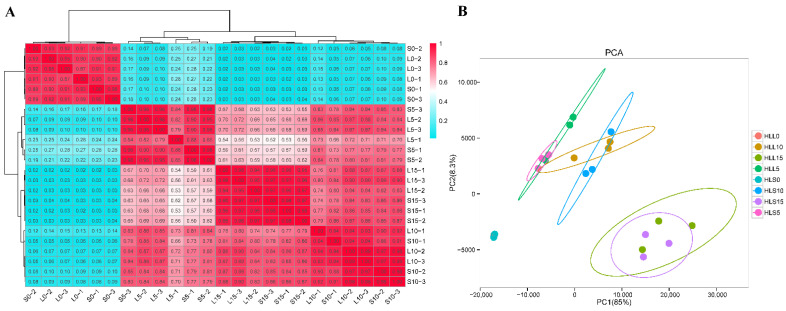
Relationship between 24 fiber samples. (**A**) Pearson’s correlation coefficient and clustering heatmap. The abscissa and ordinate are the number of samples; the order is determined by the correlation clustering results, and the color reflects the correlation between samples. (**B**) Principal component analysis of identified genes. L represents an equal mix of HL-9, HL-62, and HL-194 RNA samples; S represents an equal mix of HL-34, HL-78 and HL-159 RNA samples; 0 means 0 days after flowering; values 1, 2, 3 represent different biological replicates.

**Figure 2 genes-13-00954-f002:**
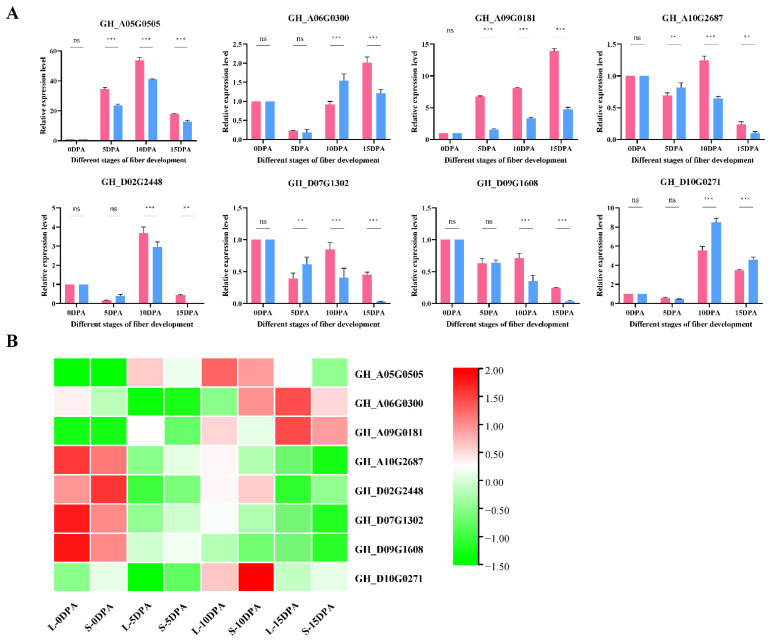
Confirmation of RNA-seq results using qRT–PCR experiments. (**A**) qRT–PCR results of 8 genes. With associated significance levels (*p* values) added above each histogram. ns indicates no significant effect at the *p* < 0.05 level, and ** and *** represent significant effect at the *p* < 0.01 and *p* < 0.001 levels, respectively. Pink bars, L; blue bars, S. (**B**) RNA-seq heatmap of 8 genes.

**Figure 3 genes-13-00954-f003:**
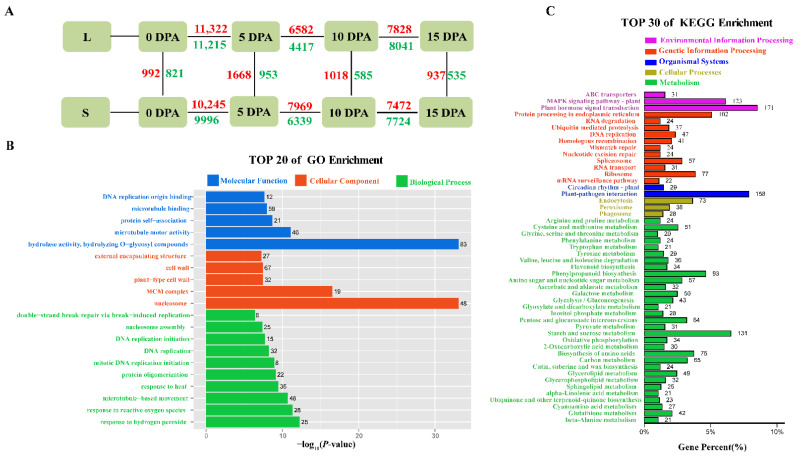
Statistical and enrichment analysis of DEGs. (**A**) Multiple comparisons of L and S in fiber development. Numbers indicate the number of DEGs. Red, upregulated; green, downregulated. (**B**) GO-term enrichment analysis of 6055 genes. (**C**) KEGG enrichment analysis of 6055 genes.

**Figure 4 genes-13-00954-f004:**
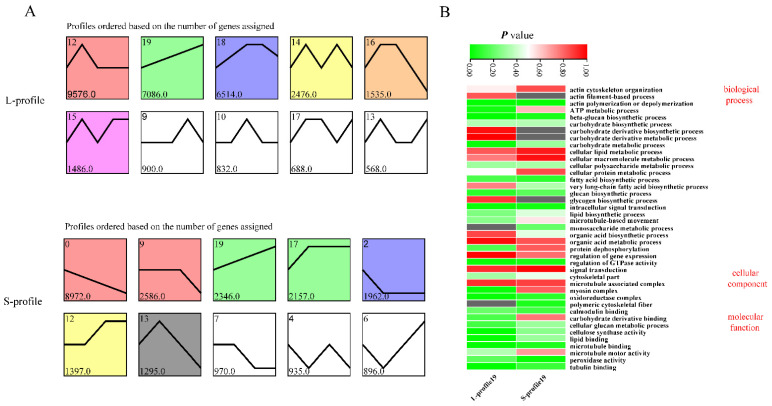
Gene expression patterns of L and S and GO enrichment of profile 19. (**A**) Gene expression patterns for the four developmental stages of L and S as inferred from STEM analysis. Each square represents the expression trend, and the text indicates the number of genes and IDs contained in the profile. (**B**) GO enrichment of L and S in Profile 19. The *p* value indicates the significance of GO terms. Gray represents no enrichment.

**Figure 5 genes-13-00954-f005:**
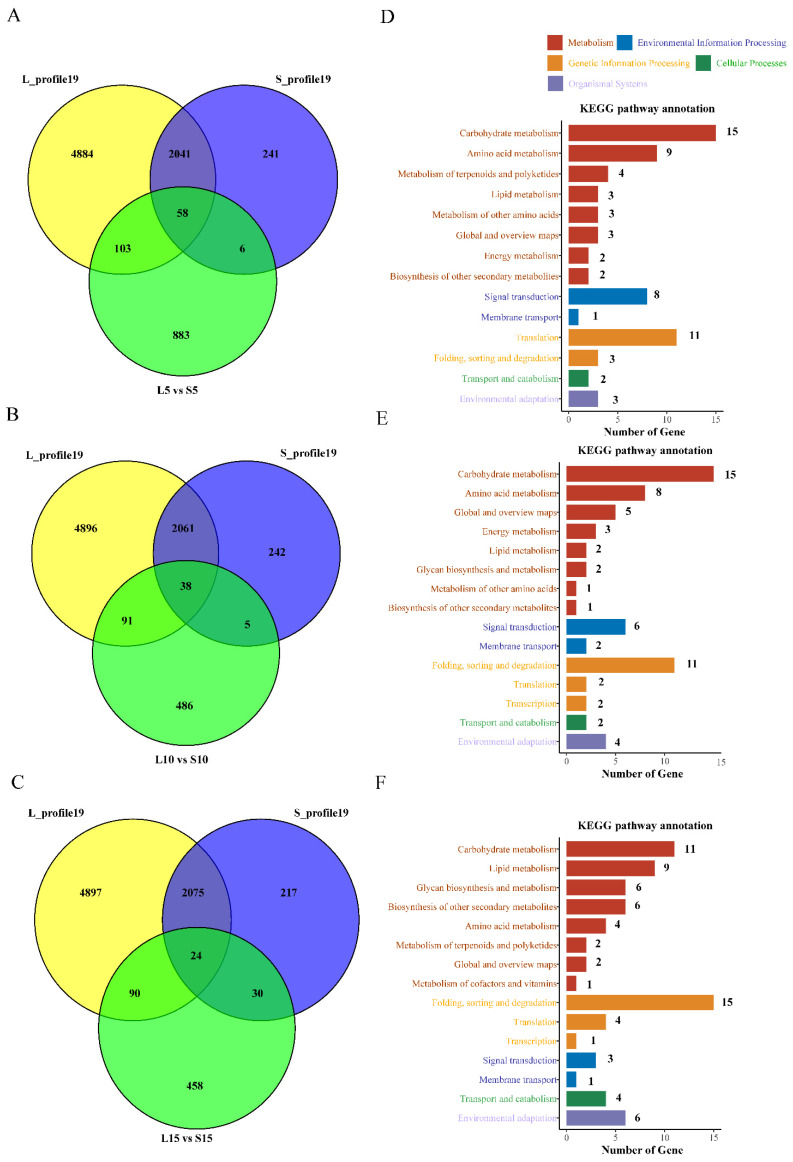
Venn diagram and KEGG pathway annotation of DEGs. (**A**) Number of DEGs shared by L and S in profile 19 and 5 DPA comparisons. (**B**) Number of DEGs shared by L and S in profile 19 and 10 DPA comparisons. (**C**) Number of DEGs shared by L and S in profile 19 and 15 DPA comparison. (**D**) KEGG pathway annotations for the 167 DEGs in A. (**E**) KEGG pathway annotations for the 138 DEGs in B. (**F**) KEGG pathway annotations for the 144 DEGs in C.

**Figure 6 genes-13-00954-f006:**
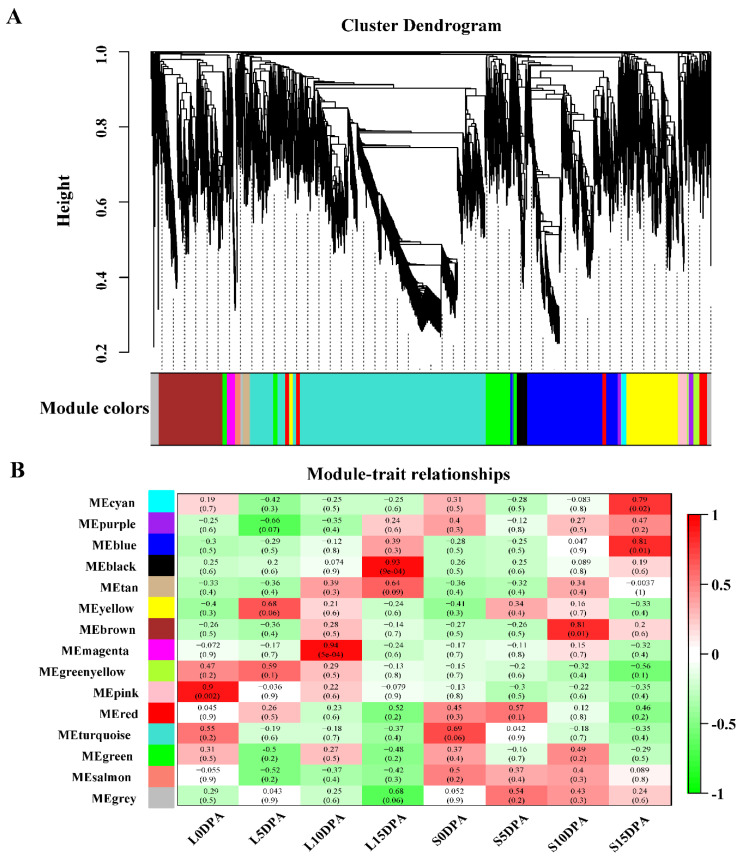
Weighted gene correlation network analysis. (**A**) Hierarchical tree diagram representing the WGCNA identification module. (**B**) Correlation diagram between modules and samples. The horizontal axis represents samples at different fiber stages, and the vertical axis represents modules. The numbers in the squares represent the correlation coefficients and *p* values between modules and traits.

**Figure 7 genes-13-00954-f007:**
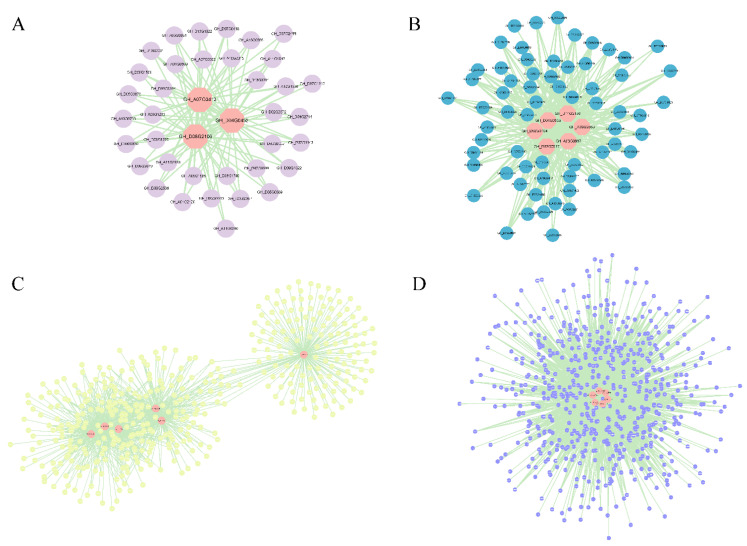
Coexpression network analysis of the (**A**) magenta module, (**B**) black module, (**C**) brown module, and (**D**) blue module. The central hub gene is represented by an orange circle.

**Table 1 genes-13-00954-t001:** Parental and extreme offspring fiber quality trait data.

Materials	Grouping	Fiber Length (mm)	Fiber Strength (cN/tex)	Micronaire (Unit)	Uniformity (%)	Elongation
Line 9	P1	28.67 ± 0.27 ^c^	30.28 ± 0.21 ^de^	4.69 ± 0.08 ^ab^	84.62 ± 0.20 ^bc^	4.33 ± 0.64 ^a^
Xinhai16	P2	35.20 ± 0.11 ^a^	49.45 ± 2.13 ^a^	4.05 ± 0.28 ^bc^	86.98 ± 0.82 ^a^	2.94 ± 0.66 ^bc^
HL-9	L	30.40 ± 0.15 ^bc^	34.59 ± 0.75 ^bc^	4.10 ± 0.19 ^bc^	85.43 ± 0.42 ^ab^	2.38 ± 0.15 ^c^
HL-62		30.16 ± 2.27 ^bc^	30.14 ± 0.49 ^de^	3.80 ± 0.48 ^c^	84.40 ± 2.03 ^bc^	3.56 ± 0.80 ^ab^
HL-194		31.62 ± 0.52 ^b^	35.51 ± 1.72 ^b^	3.60 ± 0.22 ^c^	86.35 ± 0.39 ^ab^	3.26 ± 0.74 ^abc^
HL-34	S	25.60 ± 0.30 ^d^	31.89 ± 0.62 ^cd^	4.60 ± 0.16 ^ab^	82.18 ± 0.43 ^d^	4.12 ± 0.28 ^a^
HL-78		24.68 ± 1.16 ^d^	28.37 ± 2.81 ^ef^	5.31 ± 0.78 ^a^	81.96 ± 0.90 ^d^	3.96 ± 0.83 ^ab^
HL-159		25.86 ± 0.78 ^d^	26.98 ± 1.84 ^f^	4.96 ± 0.42 ^a^	83.18 ± 1.75 ^cd^	2.95 ± 0.30 ^bc^

Different lowercase letters in each column represent significant differences at the *p* < 0.05 level (*t* test). Data in the table are the mean ± SD.

**Table 2 genes-13-00954-t002:** Hub genes and annotations.

Module	Gene ID	Arabidopsis ID	Degree	Predicted Functions (Component)
magenta	GH_A07G0413	AT4G38700	38	Involved in lignification (indicator protein)
	GH_D08G2106	AT4G23365	37	Involved in cell wall relaxation (pectinase)
	GH_D04G0452	AT5G39280	36	Involved in cell wall relaxation (expansin)
black	GH_A03G0667	AT5G50710	66	Involved in cell wall skeleton regulation (actin)
	GH_D11G2156	AT4G37630	66	Involved in microtubule formation (cyclin)
	GH_A09G2563	AT5G49360	64	Involved in secondary cell wall thickening (glycoside hydrolase)
	GH_D02G0917	AT5G24080	64	Involved in cytoskeletal rearrangement (receptor protein kinase)
	GH_D04G0535	AT3G28345	64	Involved in the differentiation of microtubule bundles (ABC transporter)
	GH_D02G0104	AT3G59690	63	Involvement in secondary wall deposition by regulating microtubules (IQ-calmodulin)
brown	GH_A05G0924	AT1G07400	170	Involved in cytoskeletal structure (heat shock protein)
	GH_D02G0408	AT4G24000	165	Involved in the hemicellulose backbone (Cellulose synthase)
	GH_A13G2053	AT1G09380	159	Involved in pectinase, cellulose regulation (transporter)
	GH_D09G2382	AT1G55850	156	Involved in the hemicellulose backbone (Cellulose synthase)
	GH_D03G0387	AT1G22360	153	Involved in glucose catalysis (glucotransferase)
	GH_D12G0897	AT4G37790	152	Involved in transcriptional regulation (homeodomain leucine zipper protein)
blue	GH_D12G1588	AT1G03870	479	Involved in cell wall adhesion (Fasciclin-like arabinogalactan protein 9)
	GH_A10G2524	AT2G39890	454	involved in cell wall proteins (proline protein)
	GH_A10G2000	AT5G12380	435	Involved in calcium channel formation (annexin)
	GH_D11G1764	AT1G66880	429	Involved in cell wall pectin binding (receptor kinase)
	GH_A07G0137	AT4G29810	425	Involved in hormone signal transduction (mitogen-activated protein kinase)
	GH_D06G1380	AT5G49720	423	Involved in cell wall assembly and xylem formation (endoglucanase)

## Data Availability

All relevant data are within the paper and its Appendix A.

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
