# Peer review of "Revealing Genetic Differences in Fiber Elongation between the Offspring of Sea Island Cotton and Upland Cotton Backcross Populations Based on Transcriptome and Weighted Gene Coexpression Networks"

_genes, 2022, doi:10.3390/genes13060954_

Round 1

Reviewer 1 Report

The main aim of the paper is to analyse the mechanism of fiber elongation in cotton in a backcross population.

The paper shows interesting information and well explained except some aspects in material and methods and the results

Specific comments

Line 82. No any study in cotton?

Like this one https://www.nature.com/articles/s41467-019-12575-x

Or this one https://journals.plos.org/plosone/article?id=10.1371/journal.pone.0082634

And this one related to fiber length

https://www.frontiersin.org/articles/10.3389/fpls.2021.796722/full

https://bmcgenomics.biomedcentral.com/articles/10.1186/s12864-019-5986-5

Line 106 From how many plants? How were plants grown? Which conditions? Field distribution?

Line 114. From how many plants per genotype?

Line 120. How many cotton bolls? Twenty per genotype? 20 in each date and genotype? How many samples in total?

Line 128. Why were samples mixed? It could allow determine specific differences of each genotype. Repetitions? Further explained in the discussion and results, but a short explanation can be given here.

Line 147. 24 cDNA libraries from which samples?

Line 199. Which test was used to set the differences?

Line 232. Improve quality of the image. Triangles do no appear in the legend of figure 1B or it cannot be seen clearly.

Line 503. Further studies can be explained here to check the importance of those genes with more experiments like genetic modified plants to overexpress those genes for example.

Author Response

Dear Professor: Thank you very much for your comments on the manuscript. We have taken your suggestions in consideration point by point and made the updating of this revision.

Introduction

Point 1: Line 82: No any study in cotton?

Response 1: Thank you so much for your suggestion. We have added literature on transcriptome studies about cotton in the manuscript.

Materials and Methods

Point 2: Line 106: From how many plants? How were plants grown? Which conditions? Field distribution?

Response 2: Thank you for reminding. A randomized zonal design with a six-row planted pattern with two rows per offspring, 2 m row length, 0.1 m plant spaced, 0.7 m row spaced, 2.35 m sowed width, three replications(50 plants per replicate), drip fertilization beneath mulched film was used for plant growth and conventional field management was used, We adjusted in the revised manuscript.

Point 3: Line 114: From how many plants per genotype?

Response 3:Thanks for your suggestion. In early October, 20 cotton bolls with natural bolls were harvested manually (one boll per plant). We added the information in the revised manuscript.

Point 4: Line 120: How many cotton bolls? Twenty per genotype? 20 in each date and genotype? How many samples in total?

Response 4: Thank you so much for your suggestion. Sampling began on July 5, 2020, with the initial date of collection being the day of flowering, labeled 0 DPA, and 36 developing cotton bolls from six offspring labeled 0 DPA, 5 DPA, 10 DPA, and 15 DPA were harvested daily at 10:00 a.m. after flowering of each replicate, then immediately placed in foam boxes with ice packs [82]. Three replicates were labeled and sampled for a total of 108 bolls. Fiber samples were placed in liquid nitrogen storage for subsequent RNA-seq and qRT–PCR validation. We adjusted in the revised manuscript and added citation.

Point 5: Line 128: Why were samples mixed? It could allow determine specific differences of each genotype. Repetitions? Further explained in the discussion and results, but a short explanation can be given here.

Response 5: Thank you for your very meaningful question. The sea island cotton and upland cotton backcross populations are more consistent in the research background. In order to reveal the overall genetic characteristics, we chose a mixed sample, which was designed to highlight the differences between the 2 groups of cotton and between the different stages of fiber development. We added a reason for choice in the revised manuscript.

Point 6: Line 147: 24 cDNA libraries from which samples?

Response 6: This is a very useful suggestion. We have combined text and tables to make a clearer presentation of the sample mixing scheme for transcriptome sequencing, and we added this part of the detailed information in the supplementary file.

Results

Point 7: Line 199: Which test was used to set the differences?

Response 7: Thank you so much for your suggestion. We modified in the manuscript.

Discussion

Point 8: Line 503: Further studies can be explained here to check the importance of those genes with more experiments like genetic modified plants to overexpress those genes for example.

Response 8: Thank you this is a very good advice. We have added the outlook of these 19 genes in the manuscript.

Figures

Point 9: Line 232: Improve quality of the image. Triangles do no appear in the legend of figure 1B or it cannot be seen clearly.

Response 9: We highly appreciate your suggestion. We have redone the PCA to make the picture clearer. We made modified and updated in the revised manuscript.

Thanks a lot for your patience and consideration.

Sincerely yours,

Wenwei Gao, PhD

Professor

Postal address: College of Agriculture, Xinjiang Agricultural University, 311 Nongda East Road, Urumqi, 830052, China.

Telephone number: 18599079231

Fax number: 0991-8762263

E-mail address: [email protected]

Reviewer 2 Report

Please check the PDF attachment.

Author Response

Dear Professor: Thank you very much for your comments on the manuscript. We have taken your suggestions in consideration point by point and made the updating of this revision.

Introduction

Point 1: Line 41: Remove“Gossypium”.

Response 1: Thanks for your suggestion. We have removed multiple recurrences of“Gossypium”in the manuscript.

Point 2: Line 46: Replace the word “low” with “lower”.

Response 2: Thank you very much for your suggestion. We had already replaced in the revised manuscript.

Point 3: Line 55: Is the 0 the fertilization day? what does the minus sign mean?

Response 3: Thanks for your question. We have checked and revised to 0 days in the manuscript.

Point 4: Line 87: “white in full if not previously mentioned”.

Response 4: Thank you so much for your suggestion. The application of WGCNA on the crop and the role, they were added in the manuscript.

Materials and Methods

Point 5: Line 100: Interspecific between which lines of G hirsutum and G barbadense ?

Response 5: Thank you for your very meaningful question. The sea island cotton and upland cotton backcross populations do pay more attention to this aspect. We currently do not know the size of the introgressed fragment. We are currently preparing for re-sequencing, and the follow-up work will also start the research on the introgressed fragment.

Point 6: Line 116: I or 1?

Response 6: Thank you for reminding. HVI is the abbreviation of English High VoIume lnstrument, We adjusted in the revised manuscript.

Point 7: Line 125: Replace the word “by Qubit 3.0” with “according to Qubit quantification”. .

Response 7: Thank you so much for your suggestion. We had already adjusted in the revised manuscript.

Point 8: Line 131: something seems to be wrong, same volumes, quantity.

Response 8: Thank you so much for your suggestion. We modified in the manuscript.

Results

Point 9: Line 198: Give the full name of FL, FS, FM, FU, FE.

Response 9: This is a very good suggestion. This makes the table contents clearer. We added the full name to the manuscript.

Thanks a lot for your patience and consideration.

Sincerely yours,

Wenwei Gao, PhD

Professor

Postal address: College of Agriculture, Xinjiang Agricultural University, 311 Nongda East Road, Urumqi, 830052, China.

Telephone number: 18599079231

Fax number: 0991-8762263

E-mail address: [email protected]

Reviewer 3 Report

I suggest improvement in the title. Authors can add something like "unrevealing Differences in Fiber Elongation Mechanisms Between the Off
spring of Sea Island Cotton and Upland Cotton Backcross Populations.  

In the abstract - "In addition, WGCNA revealed four important modules closely associated with fiber development samples, in which a total of 21 hub genes were identified, encoding 12 proteins and 7 enzymes, mainly involving heat shock protein, ABC transporter protein, mitogen-activated protein kinase, glycosyltransferase, hydrolase, and cellulose synthase" - this is too descriptive and do not bring any conclusive information.

The article is very well-drafted and easy to understand. But the major concern is the outcome of the study and the use of generated information. The number of differential expressed genes is very high. Studying over 6000 genes is very difficult and misleading. I suggest the use of publically available QTL information to prioritize the candidate genes. 

Authors can collocate the differential expressed genes with QTL hotspots. Secondly, I suggest sequencing the prioritized candidate genes to identify regulatory genetic variation. 

Authors can consider the sequencing of promoter and ORF of candidate genes with sangers sequencing or simply use the contrasting bulk for whole-genome sequencing. 

Author Response

Response to Reviewer 3 Comments

Dear Professor: Thank you very much for your comments on the manuscript. We have taken your suggestions in consideration point by point and made the updating of this revision.

Point 1: I suggest improvement in the title. Authors can add something like "unrevealing Differences in Fiber Elongation Mechanisms Between the Offspring of Sea Island Cotton and Upland Cotton Backcross Populations.

Response 1: We highly appreciate your suggestion. In this study, In this study, we performed a comprehensive analysis using RNA-Seq data from two selected groups of fiber length (long-fiber group L and short-fiber group S) at different fiber development stages in backcross populations of Sea Island and Land cotton. Enrichment revealed that DEGs involved in hormone signaling, MAPK signaling, and starch and sucrose metabolism pathways may play an important role in fiber elongation, and WGCNA-based screening identified 21 Hub genes with differential bases in land and island cotton that may be good candidates for improving cotton fiber elongation. We revised the title of the manuscript as “Revealing Genetic Differences in Fiber Elongation Between the Offspring of Sea Island Cotton and Upland Cotton Backcross Populations Based on Transcriptome and Weighted Gene Coexpression Networks”

Point 2: "In addition, WGCNA revealed four important modules closely associated with fiber development samples, in which a total of 21 hub genes were identified, encoding 12 proteins and 7 enzymes, mainly involving heat shock protein, ABC transporter protein, mitogen-activated protein kinase, glycosyltransferase, hydrolase, and cellulose synthase" - this is too descriptive and do not bring any conclusive information..

Response 2: Thank you very much for your suggestion. Analysis of the WGCNA module uncovered 21 genes closely related to fiber development, mainly involved in functions such as cell wall relaxation, microtubule formation, and cytoskeletal structure of the cell wall. We had already replaced in the revised manuscript.

Point 3: The article is very well-drafted and easy to understand. But the major concern is the outcome of the study and the use of generated information. The number of differential expressed genes is very high. Studying over 6000 genes is very difficult and misleading. I suggest the use of publically available QTL information to prioritize the candidate genes.

Response 3: Thank you so much for your suggestion. The 6000 genes were indeed too large. Considering that most of the QTL in the existing database originated from upland cotton or sea island cotton populations, which were different from the sea island cotton and upland cotton backcross populations in this study, we further used WGCNA to uncover highly related gene sets as well as explore hub genes in the network in order to explore candidate genes in a more targeted manner, and finally screened 21 genes that were considered to be key genes for fiber development.

Point 4: Authors can collocate the differential expressed genes with QTL hotspots. Secondly, I suggest sequencing the prioritized candidate genes to identify regulatory genetic variation. Authors can consider the sequencing of promoter and ORF of candidate genes with sangers sequencing or simply use the contrasting bulk for whole-genome sequencing.

Response 4: Thank you for your suggestion. Through the public database, it was found that 19 genes have amino acid sequence differences between sea island cotton and upland cotton. In fact, we are carrying out the sequencing of related genes. We plan to further screen 1-2 key genes through functional verification, and carry out the molecular mechanism of related genes. and regulatory network research, in order to reveal the genetic variation mechanism of cotton fiber development.

Thanks a lot for your patience and consideration.

Sincerely yours,

Wenwei Gao, PhD

Professor

Postal address: College of Agriculture, Xinjiang Agricultural University, 311 Nongda East Road, Urumqi, 830052, China.

Telephone number: 18599079231

Fax number: 0991-8762263

E-mail address: [email protected]
